# Topological chiral modes in random scattering networks

**Pierre A. L. Delplace**<sup>⋆</sup>

Univ Lyon, Ens de Lyon, Univ Claude Bernard, CNRS,
Laboratoire de Physique, F-69342 Lyon, France

⋆ pierre.delplace@ens-lyon.fr

## Abstract

Using elementary graph theory, we show the existence of interface chiral modes in random oriented scattering networks and discuss their topological nature. For particular regular networks (e.g. L-lattice, Kagome and triangular networks), an explicit mapping with time-periodically driven (Floquet) tight-binding models is found. In that case, the interface chiral modes are identified as the celebrated anomalous edge states of Floquet topological insulators and their existence is enforced by a symmetry imposed by the associated network. This work thus generalizes these anomalous chiral states beyond Floquet systems, to a class of discrete-time dynamical systems where a periodic driving in time is not required.



# 1 Introduction

Chiral edge states constitute a ubiquitous signature of the topological properties of many physical systems. The emergence of such one-way dissipationless modes is well understood in condensed matter with the celebrated bulk-boundary correspondence [1–7]. According to this correspondence, the number of chiral edge modes of a two-dimensional gapped system is fixed by the value of a topological index, namely the first Chern number, that characterizes the topological property of the bulk eigenstates parametrized over the Brillouin zone. While the demonstration of the topological origin of the quantized Hall conductivity launched the field of topological phases of matter [8–10], the universality of the bulk-edge correspondence allowed the spread of topological physics beyond condensed matter physics, from quantum to classical systems of various types. This recent expansion was made possible in particular thanks to the versatility and the high degree of control of various artificial crystals or meta-materials in which chiral boundary modes can be directly probed experimentally [11–17].

Remarkably, the bulk-edge correspondence was recently jostled in periodically driven quantum systems, also called Floquet systems, in which robust chiral boundary modes may emerge while the Chern numbers vanish [18]. These unexpected *anomalous* chiral edge states, that have no counterpart in static systems, motivated Rudner, Lindner, Berg and Levin to propose a new bulk-boundary correspondence for periodically driven systems [18], later formally demonstrated [19], that correctly predicts these new topological boundary modes, but where the seminal Chern numbers are unseated and replaced by more accurate topological indices. The existence of these anomalous chiral states have been confirmed since by direct observations in various photonic devices [20–22]. The power of this breakthrough is that every chiral edge states are treated on the same footing, whether or not the topological state is anomalous. But on the same time, it does not help to understand what is specific to anomalous chiral boundary states neither to engineer one on purpose.

The present work answers these questions beyond the context of Floquet physics, by considering unitary discrete-time dynamics that are not necessarily periodic in time. The emergence of anomalous chiral modes is found to be a generic topological property of arbitrary oriented scattering networks whose meaning can be twofold: they can either represent a discrete-time

Hamiltonian dynamics, or constitute the actual physical systems through which wave packets propagate. As a result, anomalous chiral modes are not specific to periodically driven (Floquet) systems. More generally, we show how topological properties of a coherent (or quantum) state evolving in a network can simply be inferred from the properties of the network itself.

Discrete-time dynamics are abundantly used in Floquet topological physics, not only theoretically where they were first introduced to illustrate the anomalous regime [18, 23] and its physical consequences [24, 25], but also experimentally in classical optics [20, 21, 26] as well as in quantum optics [27–29] and cold atoms physics [30, 31] where they are often referred to as discrete-time quantum walks [30, 32, 33]. Another interest of discrete-time dynamics is that they constitute the building blocks of arbitrary dynamics, since the evolution operator of a continuous dynamics can be decomposed as an infinite product of infinitesimal free evolutions governed by instantaneous Hamiltonians taken at successive times. Since these Hamiltonians generically do not commute, the evolution operator can be cumbersome to manipulate. It can thus be convenient to approximate the dynamics by a finite sequence of unitary operations, as it is currently done numerically to compute the evolution operator.

This paper is organized as follows: In section 2, we establish a direct mapping between oriented scattering networks and discrete-time periodic tight-binding models introduced in the context of Floquet anomalous topological phases [18, 23]. This allows us to *represent* periodically driven tight-binding models by networks in which we unveil a *phase rotation symmetry* [34] that imposes the existence anomalous topological chiral states (section 3). Remarkably, the topological properties of oriented networks one ends up with were discussed independently in the literature [16, 34–38] ; these mappings thus clarify the similarities between their topological properties and those of Floquet tight-binding models [18, 23]. This simple construction also allows us to propose other Floquet tight-binding models that host anomalous topological states. Furthermore, and more importantly, the interpretation of the phase rotation symmetry as a simple generic property of the network allows us to generalize it to arbitrary scattering networks and thus to investigate the topological properties of discrete-time dynamics beyond Floquet systems. This case is addressed in section 4 by means of graph theory applied to random networks that preserve unitarity. The existence of chiral modes that generalize the anomalous ones in the Floquet case is inferred from simple graphical properties of the graphs. We discuss the topological nature of this information in section 5 and confirm it by computing the flow induced by the dynamics in the random graph that is found to be quantized in agreement with the graph theory approach.

## 2 Discrete-time dependent tight-binding models as topological cyclic oriented scattering networks

### 2.1 Mapping of a canonical Floquet tight-binding model onto the L-lattice model

Discrete-time dependent tight-binding models were proposed on the honeycomb lattice [23] and the square lattice [18] to realize Floquet "topological insulators". In these models, the hopping energy terms $J$ that couple nearest neighbors are successively switched on and off around each plaquette, therefore inducing a breaking of time-reversal symmetry in the periodic dynamics. Each time period $T$ is thus decomposed in a finite time-ordered sequence of constant in time tight-binding Hamiltonians that describe arrays of uncoupled dimers whose sites are

coupled by $J_i$ during a time $\tau_i$. The bulk (Bloch) Hamiltonian thus reads

$$H(t, \mathbf{k}) = \begin{cases} J_1 h_1(\mathbf{k}) & t_0 = 0 < t < t_1 \\ J_2 h_2(\mathbf{k}) & t_1 < t < t_2 \\ \dots & \\ J_n h_n(\mathbf{k}) & t_{n-1} < t < t_n = T \end{cases}, \tag{1}$$

where $\mathbf{k}$ is the quasi-momentum and $h_j$ is an hermitian matrix. Setting the dimensionless *phase coupling parameters* $\theta_j \equiv J_j \tau_j / \hbar$ where the duration $\tau_j = t_j - t_{j-1}$, an evolution operator $U_j(\mathbf{k}) = e^{-i\theta_j h_j(\mathbf{k})}$ can be assigned to each time step. These discrete-time dependent tight-binding models have a scattering network representation that we present now.

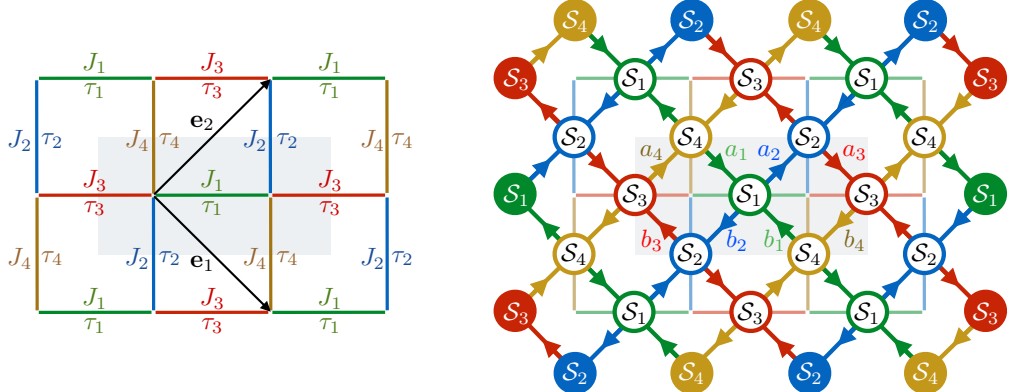

Figure 1: (left) Discrete-time dependent tight-binding model for the square lattice and (right) its scattering network representation (L-lattice). The successive time steps of duration $\tau_i$ with couplings of amplitude $J_i$ are represented in the same color than the scattering matrices they generate. A unit cell of the two lattices is represented by a grey rectangle, and a basis of the Bravais lattice is given by $\vec{e}_1$ and $\vec{e}_2$. Note that while the tight-binding model has two inequivalent sites per unit cell, the L-lattice has eight inequivalent oriented links (and four scattering matrices).

As a canonical example, let us start with the square lattice, depicted in figure 1 (a), since it is certainly the most discussed in the literature. In that case,

$$h_1(\mathbf{k}) = \begin{pmatrix} 0 & e^{ik_+} \\ e^{-ik_+} & 0 \end{pmatrix}, \quad h_2(\mathbf{k}) = \begin{pmatrix} 0 & e^{ik_-} \\ e^{-ik_-} & 0 \end{pmatrix},$$
$$h_3(\mathbf{k}) = \begin{pmatrix} 0 & e^{-ik_+} \\ e^{ik_+} & 0 \end{pmatrix}, \quad h_4(\mathbf{k}) = \begin{pmatrix} 0 & e^{-ik_-} \\ e^{ik_-} & 0 \end{pmatrix}, \tag{2}$$

with $k_\pm = \mathbf{k} \cdot \frac{\mathbf{e}_1 \pm \mathbf{e}_2}{2}$. The evolution operators assigned to the successive time-step can be factorized as

$$\begin{aligned} U_1(\mathbf{k}) &= \mathcal{B}(k_+)S(\theta_1)\mathcal{B}(k_+), & U_2(\mathbf{k}) &= \mathcal{B}(k_-)S(\theta_2)\mathcal{B}(k_-), \\ U_3(\mathbf{k}) &= \mathcal{B}(-k_+)S(\theta_3)\mathcal{B}(-k_+), & U_4(\mathbf{k}) &= \mathcal{B}(-k_-)S(\theta_4)\mathcal{B}(-k_-), \end{aligned} \tag{3}$$

where

$$\mathcal{B}(k) = \begin{pmatrix} 0 & e^{i\frac{k}{2}} \\ e^{-i\frac{k}{2}} & 0 \end{pmatrix}, \quad S_j = \begin{pmatrix} \cos\theta_j & -i\sin\theta_j \\ -i\sin\theta_j & \cos\theta_j \end{pmatrix} \tag{4}$$

so that each part of the evolution can be split into elementary steps that encode either a displacement through $\mathcal{B}(k)$ or a coupling through the scattering matrix $S_j$. The (Floquet) evolution operator after a period from $t = 0$ to $t = T$, reads

$$U_F(\mathbf{k}) = U_4(\mathbf{k})U_3(\mathbf{k})U_2(\mathbf{k})U_1(\mathbf{k}). \tag{5}$$

Substituting the $U_j$'s by their decomposition (3) yields

$$U_F(\mathbf{k}) = \mathcal{B}(-k_-) S_4 T_{\mathbf{e}_2} S_3 T_{-\mathbf{e}_1} S_2 T_{-\mathbf{e}_2} S_1 \mathcal{B}^\dagger(k_+), \tag{6}$$

where $T_{\mathbf{e}_j}$ is a "sublattice dependent" translation operator in the directions $\pm\mathbf{e}_j/2$ that reads

$$T_{\mathbf{e}_j} \equiv \begin{pmatrix} e^{i\frac{\mathbf{k}.\mathbf{e}_j}{2}} & 0 \\ 0 & e^{-i\frac{\mathbf{k}.\mathbf{e}_j}{2}} \end{pmatrix}. \tag{7}$$

The definition of the Floquet evolution operator is not unique, as it depends on the choice of origin of time. One can thus choose a different origin by a cyclic permutation of the unitary steps in (5). Starting the evolution by the translation operation $T_{\mathbf{e}_2}$, one thus equivalently describes the periodic dynamics by the Floquet operator

$$\tilde{U}_F(\mathbf{k}) = S_4 T_{\mathbf{e}_2} S_3 T_{-\mathbf{e}_1} S_2 T_{-\mathbf{e}_2} S_1 T_{\mathbf{e}_1}. \tag{8}$$

The expression (8) has a straightforward interpretation in real space: it shows that the evolution can be seen as a staggered succession of local scattering processes $S_j$ followed by a free propagation in the directions $\pm\mathbf{e}_1$ or $\pm\mathbf{e}_2$. Representing the scattering processes by circles, and the free propagations by straight arrows in directions $\pm\mathbf{e}_j$, one gets a version of the celebrated scattering matrix network called *L-lattice* that was introduced by Chalker and Coddington in the context of the quantum percolation transition between plateaus of the quantized Hall effect (see figure 1 (b)) [39,40]. The corresponding Floquet tight-binding model is superimposed to this network in figure 1 (b) to emphasize the connection between the two models. The construction of the oriented scattering network from the discrete-time dependent tight-binding model becomes intuitive: the pairs of sites that are coupled during a time-step in the tight-binding model are replaced by scattering matrices. These matrices are then connected by oriented links whose orientation satisfies the time-ordering of the dynamics. The oriented scattering lattice then obtained thus represents a periodic (Floquet) dynamics. This periodicity is visible directly on the network since any path along the oriented links encounters an ordered periodic sequence of scattering events. For this reason, such networks can be qualified as *cyclic* [34]. Note that a unit cell of this oriented lattice contains 4 scattering nodes and 8 oriented links, while the original tight-binding model only contains 2 inequivalent sublattices.

Finally, alternatively to the Floquet evolution operator and following Chalker, Coddington and Ho, [39,40], one can write down a "one-step" evolution operator on the network as

$$\mathcal{U}(\mathbf{k}) \equiv \begin{pmatrix} 0 & 0 & 0 & S_4 T_{\mathbf{e}_2} \\ S_1 T_{\mathbf{e}_1} & 0 & 0 & 0 \\ 0 & S_2 T_{-\mathbf{e}_2} & 0 & 0 \\ 0 & 0 & S_3 T_{-\mathbf{e}_1} & 0 \end{pmatrix} \tag{9}$$

in the basis of the 8 oriented links $(a_1, b_1, \dots, a_4, b_4)$ where the pair $(a_j, b_j)$ enters the scattering matrix $S_j$. The form of the one-step evolution operator $\mathcal{U}$ is reminiscent of the cyclic structure of the oriented network. The different Floquet evolution operators (i.e. defined with different origins of time) are then simply inferred by taking $\mathcal{U}^4(\mathbf{k})$. A correspondence between the Floquet (or discrete-time quantum walk) point of view given by (8) and the network point of view given by (9), in particular for the characterization of their topological properties, is detailed in [34].

Note that the tight-binding model introduced in [18] that gives rise to various Floquet topological states actually slightly differs from the one introduced here in two ways: first the product $\tau_j J_j = \tau J$ is fixed to a single parameter, and second, a fifth time step is added during which the couplings $J$ are set to 0 but where a staggered on-site potential is switched on.

Three regimes were found: topologically trivial, Chern insulator, and anomalous. The version of the model introduced here exhibits the same three regimes. The anomalous one, which we focus on, is illustrated in figure 4 by its eigenvalues spectrum in a cylinder geometry, in the case where the scattering parameters are equal $\theta_1 = \theta_2 = \theta_3 = \theta_4$. In this particular case, the size of the unit cell is divided by a factor two so that the network reduces to the original L-lattice [39, 40] for which the topological properties have been investigated in [34, 35, 37, 38]. In that case, it was found that the evolution operator exhibits only two distinct topological regimes: trivial and anomalous, exactly like in the Floquet tight-binding model [18] in the absence of the fifth step.

## 2.2 Construction of other cyclic oriented scattering networks

One can now easily infer the scattering network representation of other discrete-time dependent tight-binding models. In particular, one can consider a similar periodic stepwise evolution that was originally proposed on a honeycomb lattice in a pioneering work dealing with Floquet topological states [23]. Following the same lines as in section 2, one obtains an oriented scattering Kagome network, as shown in figure 2. The topological properties of this oriented scattering network were discussed in [34, 36]. In particular it was found that this model exhibits three distinct topological gapped regimes: trivial insulating, Chern insulating and topological Floquet anomalous. Previously, the independent study of the driven tight-binding model on the honeycomb lattice [23] found the same three regimes. The coincidence of these results becomes transparent thanks to the explicit mapping between these two models.

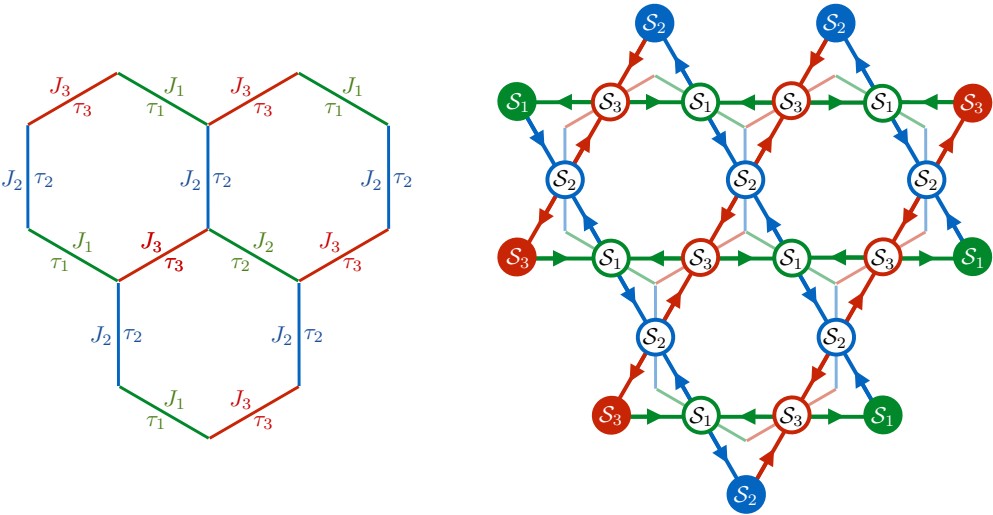

**Figure 2:** (left) Discrete-time dependent tight-binding models for the honeycomb lattice and (right) its scattering network representation (Kagome oriented lattice). The successive steps of duration $\tau_i$ and amplitudes $J_i$ are represented in the same color than the scattering matrices they generate.

One can go a step further by proposing other periodic discrete-time dependent tight-binding models together with their cyclic oriented network representation. For instance, one could think about a triangle lattice, whose coupling between nearest sites switch on and off trimers in time as depicted in figure 3 (a), where the succession in time of the different steps is chosen to induce a favored rotation that breaks time-reversal symmetry. The corresponding scattering network, shown in figure 3 (b), is an oriented triangular network.

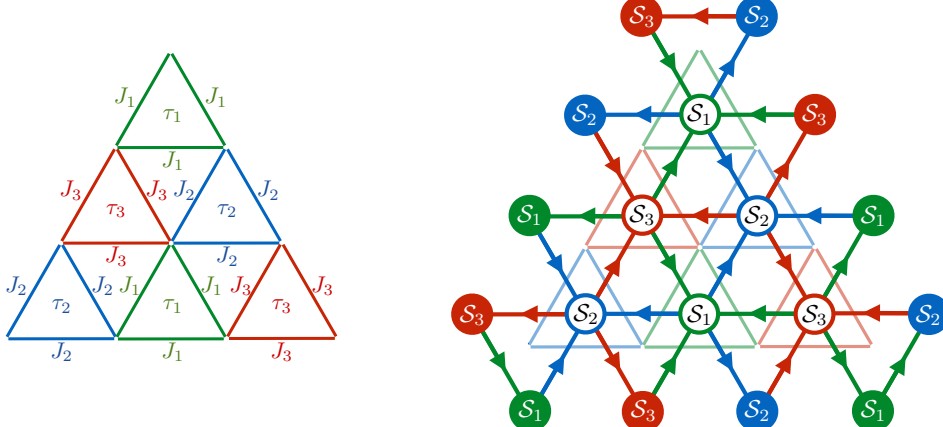

Figure 3: (left) Discrete-time dependent tight-binding models for the triangular lattice and (right) its scattering network representation (triangular oriented lattice). The successive steps of duration $\tau_i$ and amplitudes $J_i$ are represented in the same color than the scattering matrices they generate.

## 3 Strong phase rotation symmetry of cyclic oriented networks

### 3.1 Strong phase rotation symmetry

The topological properties of the evolution operator of cyclic oriented networks were detailed in [34]. In particular, it was pointed out for the L-lattice and for the Kagome lattice that, when the scattering nodes fully transmit the incoming states into one direction only, so that the lattices consist in arrays of "classical" disconnected loops, then the Floquet evolution operator acquires a (strong) *phase rotation symmetry* that constrains the Chern number of each band to vanish. It was also found that, at this special symmetry point, the quantum non-interacting spinless fermionic problem coincides with a strongly interacting classical counterpart, allowing for instance a well-approximated description of the short time quantum dynamics from the classical limit even slightly away from the symmetric point [41].

Such a symmetry is defined as follows: a bulk unitary evolution on the network characterized by the one-step operator $\mathcal{U}(\mathbf{k})$ is said to own a strong phase rotation symmetry if there exists a unitary operator $\Lambda$ such that

$$\Lambda \mathcal{U}(\mathbf{k})\Lambda^{-1} = e^{i2\pi/N}\mathcal{U}(\mathbf{k}) \tag{10}$$

where $N$ is the number of bands. Notice that, in the context of quantum dynamics in finite dimensional vector spaces, $\Lambda$ and $\mathcal{U}$ can be interpreted from (10) as two elements of the Abelian group of unitary rotations in the projective Hilbert space (or ray space) that identifies two states that only differ from each other by a $U(1)$ phase factor [42]. In that case, these elements could be interpreted as the exponential of the position and momentum operators that are represented by the Sylvester's "clock" matrix $\Lambda_0$ and the "shift" matrix $P_0$ respectively, that read

$$\Lambda_0 = \begin{pmatrix} \lambda & 0 & \cdots & & 0 \\ 0 & \lambda^2 & \cdots & & 0 \\ & & \ddots & & \vdots \\ & & & & 0 \\ 0 & \cdots & 0 & & \lambda^N \end{pmatrix} \quad P_0 = \begin{pmatrix} 0 & \cdots & \cdots & 0 & 1 \\ 1 & 0 & \cdots & & 0 \\ & 1 & & & \vdots \\ & & \ddots & & 0 \\ 0 & \cdots & 0 & 1 & 0 \end{pmatrix}, \tag{11}$$

where $\lambda = e^{i2\pi/N}$. When the values of the scattering parameters are such that a cyclic oriented network simply consists in an array of classical disconnected loops, then the Bloch evolution

operator can always be factorized in a shift operator form $\Lambda_0$ (up to a change of basis encoded though a matrix $M$). It follows (see appendix A) that the strong phase rotation symmetry (10) is automatically satisfied and that the symmetry operator is given by the clock matrix as

$$\Lambda = M^{-1}\Lambda_0 M \tag{12}$$

in the original basis of the links in which the "one-step" evolution operator $\mathcal{U}(\mathbf{k})$ is written.

This result guaranties that when the scattering parameters yield a classical configuration of disconnected loops in any cyclic oriented scattering network, then the bulk eigenstates carry a vanishing Chern number [34]. We will see in section 4 that such configurations exist in any unitary scattering network, beyond the cyclic ones.

It is worth noticing that away from this critical point, the strong phase rotation symmetry breaks down, but the Chern numbers cannot change value by virtue of their topological nature, unless a gap between eigenvalues of the evolution operator closes.[1] This is illustrated in figure 4 by the eigenphase spectra of the evolution operator of the three cyclic oriented networks discussed above in a cylinder geometry. If all the gaps eventually close simultaneously, the system undergoes a topological transition from the anomalous to the trivial regime. It was pointed out for the L-lattice that this topological transition coincides with the percolation transition in the network [35]. It follows from the mapping established above that the topological transition between Floquet anomalous and trivial regimes in discrete-time tight-binding models should coincide with a percolation transition.

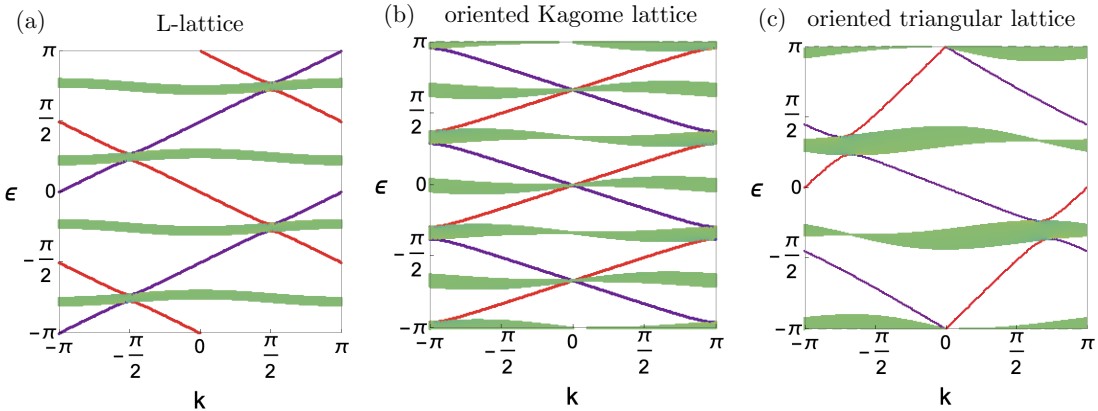

**Figure 4:** Eigenvalue phase spectrum $\epsilon$ of the "one-step" Chalker-Coddington evolution operator (9) for three cyclic oriented lattices in a cylinder geometry, slightly away from strong phase rotation symmetric point, as a function of the quasi-momentum in the direction parallel to the edges of the cylinder. The color code represent the mean position in the corresponding eigenstate, from one edge (red) to the other (blue) through the bulk (green).

## 3.2 Anomalous Floquet topological interface states

The vanishing of the Chern numbers due to the strong phase rotation symmetry does not guarantee automatically the existence of an anomalous Floquet topological regime, since it is of course also consistent with a trivial one. Besides, such an anomalous regime is actually ill-defined in a scattering network, since the bulk winding invariant (that characterizes the full evolution operator over a period) depends on the choice of the unit-cell in the bulk [34]. Accordingly, the existence of chiral boundary modes depends on the boundary conditions. This

---

[1] Note that since at the phase rotation symmetric point, the array consists in disconnected loops, no state can propagate through the system, which guarantees the existence of gaps in the eigenvalue spectrum of the evolution operator.

is clear when considering an array of disconnected loops (strong phase rotation symmetric point), as shown in figure 5; it is always possible to either add or remove an isolated boundary mode represented by a large loop that surrounds the array. This arbitrary change of boundary is made by leaving the array unchanged in the bulk, and thus preserving the bulk topological properties.

It is worth noticing that this ambiguity in the definition of the bulk topological index for the networks is fixed when mapping a discrete-time periodic tight-binding model onto a scattering network, since in that case the boundary conditions on the network are imposed and inherited from the ones of the tight-binding model. In that case, it is not allowed to modify the boundary of the network arbitrarily, and accordingly, the bulk winding number of the evolution operator is well-defined and correctly predicts the existence of chiral boundary modes. As a biproduct, scattering networks allow for more various boundary conditions than their tight-binding analogues. The reason being that the number of degrees of freedom increases when mapping the tight-binding models onto the scattering ones, as already pointed out for the mapping to the L-lattice in section 2.

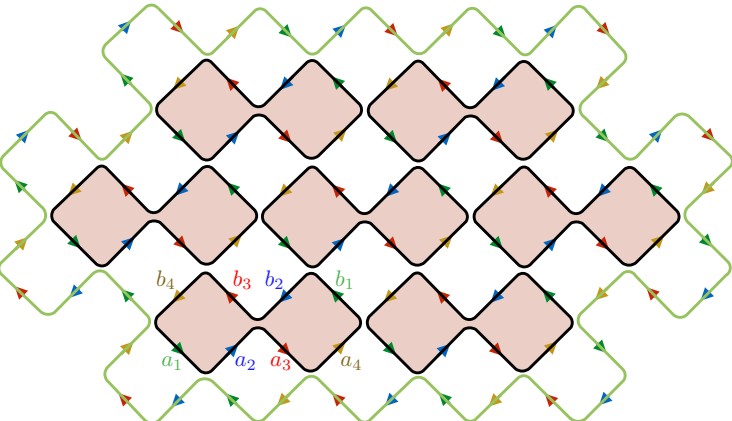

**Figure 5:** At the strong phase rotation symmetry point, an anomalous chiral boundary mode in a scattering network (in green) is disconnected from the rest of the lattice. It can thus be added or removed without modifying the bulk, by simply changing the size of the system. This is not the case for chiral boundary modes constrained by Chern numbers.

Nonetheless, it is possible to characterize the topological Floquet anomalous phases in scattering networks by using the relative bulk indices from side to side of an interface [34]. This approach predicts correctly the number of topologically protected chiral states that can propagate at the interface of two domains with different bulk topological indices. At the strong phase rotation symmetric point, this chiral mode appears explicitly visually at the frontier between two arrays of loops that circulate in opposite directions. For example, in the square network, one can consider two domains of clockwise and anti-clockwise loops, as displayed in figure 6. These two domains therefore both correspond to a vanishing Chern number regime, because of the strong phase rotation symmetry they satisfy. Concurrently, they cannot be continuously deformed one into each other. It follows that a meandering oriented path necessarily exists at the interface between the two domains. Unlike a boundary mode (see figure 5) this interface mode cannot be removed (or added) since it has to satisfy the shape of the underlying L-lattice. This is the representation of an interface anomalous Floquet state whose topological origin has been discussed for cyclic oriented networks in [34,38].

To summarize, phase rotation symmetry allows one to turn the problem of the search for anomalous Floquet chiral states into the one of the search for disconnected "meandering graphs" that emerge at the interface between domains of loops of opposite orientation. This

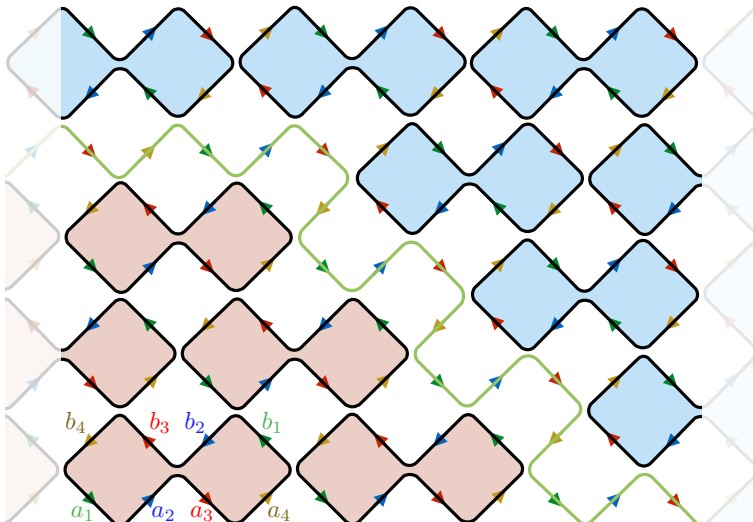

**Figure 6:** Two domains of close loops with clockwise (red) and anti-clockwise (blue) circulation. At the interface a chiral mode (green) of the Floquet anomalous regime appears.

is of great help to look now for such chiral states in more complexe systems whose discrete-time dynamics is represented by (or occurs on) random networks, since the latest problem can be addressed within elementary graph theory. In the next section, we show that the possible existence of clockwise and anti-clockwise domains of loops is inherent to arbitrary unitary scattering networks, and that their "interface" necessarily hosts an oriented meandering oriented graph, whatever their width. This generalizes the anomalous Floquet chiral states at the strong phase rotation symmetric point beyond the cyclic oriented networks that only produce Floquet dynamics.

# 4 Chiral interface states beyond periodic dynamics from graph theory

In this section, we justify geometrically the possible existence of domains of loops of opposite orientations in arbitrary *disordered* scattering networks (not necessarily periodic lattices), and show that they necessarily lead to the existence of an interface chiral state. Oriented networks models were originally introduced in physics to precisely tackle the percolation transition in disordered systems [39]. The disorder was then included through random scattering coefficients in the L-lattice model. Here we follow a different strategy, by considering arbitrary scattering networks where the connectivity of the nodes is also random. The approach is based on elementary graph theory that we succinctly remind below to the readers that are unfamiliar with these concepts [43].

## 4.1 A brief introduction to Eulerian graphs

### 4.1.1 Walk on a graph

A graph $\mathcal{G}$ is a set of vertices and of pairs of vertices called edges. To avoid any confusion with the actual edges (i.e. boundaries) of the physical system that are also discussed in the article, the edges of the graph will be referred to as *links* in the following as they connect two vertices. The graphs we are concerned with are said to be *simple* (i.e. there is at most one

link that connects two vertices), *planar* (i.e. purely two-dimensionnal), *connected* (i.e. there is no isolated vertex) and *oriented* (i.e. an orientation is assigned to each link). Physically, this orientation captures the sens of evolution of the dynamics: the component of a state on a given link at time $t$ will be scattered at the vertex pointed by the orientation of this link during the next step-time evolution $t + 1$. This succession of links and nodes during the evolution is called a *walk*.

### 4.1.2 Definition of an Eulerian graph

If there exists a closed walk that visits all the links of the graph exactly once, then the graph is called *Eulerian*, and the closed walk is called an *Eulerian circuit*. A necessary and sufficient condition for the graph to be Eulerian is that the number of links at any vertex is even. An example of such a graph is displayed in figure 7 (a). Since physically, each vertex of the graph describes a *unitary* scattering process, it follows that the number of links pointing toward a vertex equals the number of links pointing backward. As a consequence all the graphs of interest for this study are Eulerian.

### 4.1.3 Two-colour theorem

An Eulerian graph can also be thought of as a two-colored graph, meaning that exactly two colors are required to color its faces such that two adjacent faces always carry different colors. This is sometimes refered to as the *two-colour theorem*. Obviously, there is a bijective mapping between the colors of the faces of the graph and the orientation of the links delimiting these faces, so that an orientation – say $(-)$ for clockwise and $(+)$ for anti-clockwise – can be equivalently assigned to each face, as shown in figure 7 (a).

### 4.1.4 Bi-partite dual graph

It can be useful to introduce the dual graph $\mathcal{G}^*$ of $\mathcal{G}$. By construction, its vertices are given by the faces of $\mathcal{G}$, and its links relate these vertices by crossing the links of $\mathcal{G}$, as shown in figure 7 (b). Note that, according to the standard definition [43], the "exterior" of the graph is considered itself as a face, hence the existence of a blue vertex (whose position has been chosen arbitrary) that is connected to several red vertices. As a consequence, a walk on the dual graph is tantamount to a succession of "jumps" between adjacent faces of $\mathcal{G}$. It follows that $\mathcal{G}$ is Eulerian if and only if $\mathcal{G}^*$ is bipartite, i.e. the vertices of $\mathcal{G}^*$ can be colored with exactly two colors such that its links only connect two vertices of different colors. This property is used in appendix C to prove property 1 on the existence of a chiral interface mode.

### 4.1.5 Veblen decomposition and phase rotation symmetry

In the following we will use another important property, known as Veblen's theorem [44]. This theorem states that it is always possible to decompose a planar Eulerian graph as a union of disjoint simple *cycles* (i.e. closed walks containing as many edges as vortices). These cycles can simply be seen as polygons. This is the generalization of the domains of loops depicted in figures 5 and 6 where all the loops can now differ in size and shape. Physically, such a situation is again achieved when tuning the scattering parameters in a "fully reflecting" or "classical" configuration where, at each vertex, the scattering matrix has exactly one non-vanishing element of modulus 1 per line and per column. When the network is a periodic lattice, the Veblen decomposition corresponds to a strong phase rotation symmetric point as discussed in the previous sections.

For an arbitrary Eulerian graph, several such decompositions are possible. Of importance is the decomposition illustrated in figure 7 (c) where each simple cycle surrounds a face of the

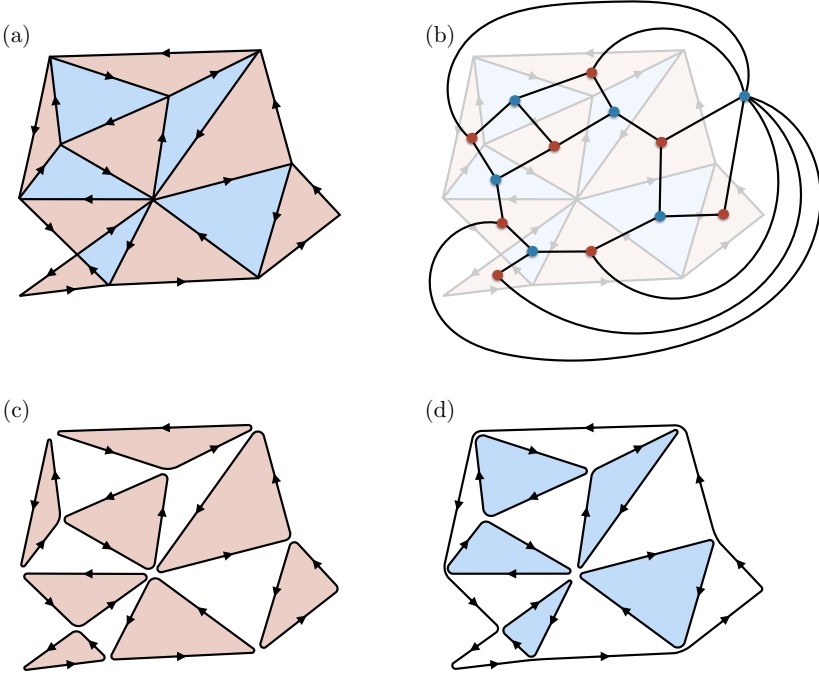

**Figure 7:** (a) Example of an oriented Eulerian graph $\mathcal{G}$. The faces have been colored according to the orientation of the links. (b) Dual graph $\mathcal{G}^*$ of $\mathcal{G}$. (c) Minimal Veblen decomposition of $\mathcal{G}$. (d) Other possible decomposition whose one of the cycle follows the boundary of the graph. Its orientation is opposite to those of the minimal cycles that surround the faces of the original graph $\mathcal{G}$, but the same as the ones of the minimal Veblen decomposition (c).

original graph. Such a decomposition is unique, and we shall refer to it as the *minimal* Veblen decomposition. Clearly, for such a decomposition, all the cycles have a well defined orientation (+ in the example of figure 7 (c)) so that there is no ambiguity to talk about the orientation of the graph itself. Interestingly, the same graph also admits another Veblen decomposition where the simple cycles surround other faces of the original graph with an opposite orientation, and where an extra cycle delimiting the boundary of the graph necessarily exists (see figure 7 (d)). These two decompositions correspond to the two strong phase rotation symmetric points discussed in the case of lattices, and the boundary cycle corresponds to the chiral edge mode of the anomalous Floquet phase. Note that this boundary cycle has an opposite orientation to those of the simple cycles that it encloses; its orientation is the one of the minimal Veblen decomposition.

## 4.2  Interface chiral Eulerian circuits

As argued in section 3 for oriented lattices, it is more meaningful to look for *interface* chiral modes than *boundary* chiral modes, since the existence of the latest depend on the boundary conditions of the graph. On the language of graphs, these interface chiral modes constitute Eulerian walks. We shall show that two domains of opposite orientation necessarily host an oriented Eulerian walk at their interface, whatever the shape and the size of the interface.

To do so, let us consider an arbitrary Eulerian graph, and let us split it into three domains nested into each other, as in figure 8 (a) and (b). The faces of the three domains have been colored with two colors to highlight that each domain remains an Eulerian graph. The separation between the domains is made by two oriented closed walks (circuits): the circuit $\mathcal{C}_1$ delimit the interior domain $D_1$ and the circuit $\mathcal{C}_2$ delimit the exterior domain $D_2$. The interface graph

left in between is noted **I**. The two circuits that surrounds it are not included in its definition: they belong to $D_2$ or $D_1$. We want to establish under which condition the interface graph **I** constitutes or hosts an interface chiral mode.

Importantly, there is a choice of orientation for each circuit. The two examples displayed in figure 8 (a) and (b) correspond to two different orientations for the larger circuit $\mathcal{C}_2$. Note that the orientation of the circuits $\mathcal{C}_1$ and $\mathcal{C}_2$ fixes the orientation of the domains $D_1$ and $D_2$ respectively, by fixing their Veblen decomposition. In particular, $D_1$ and $D_2$ have opposite orientations if and only if $\mathcal{C}_1$ and $\mathcal{C}_2$ have the same orientation. Concretely, in figure 8 (a) and (b), the domain $D_1$ has a clockwise orientation since its Veblen decomposition is made of the disconnected union of blue polygons. For the same reason, the orientation of $D_2$ is counter-clockwise in figure 8 (a) and clockwise in figure 8 (b).

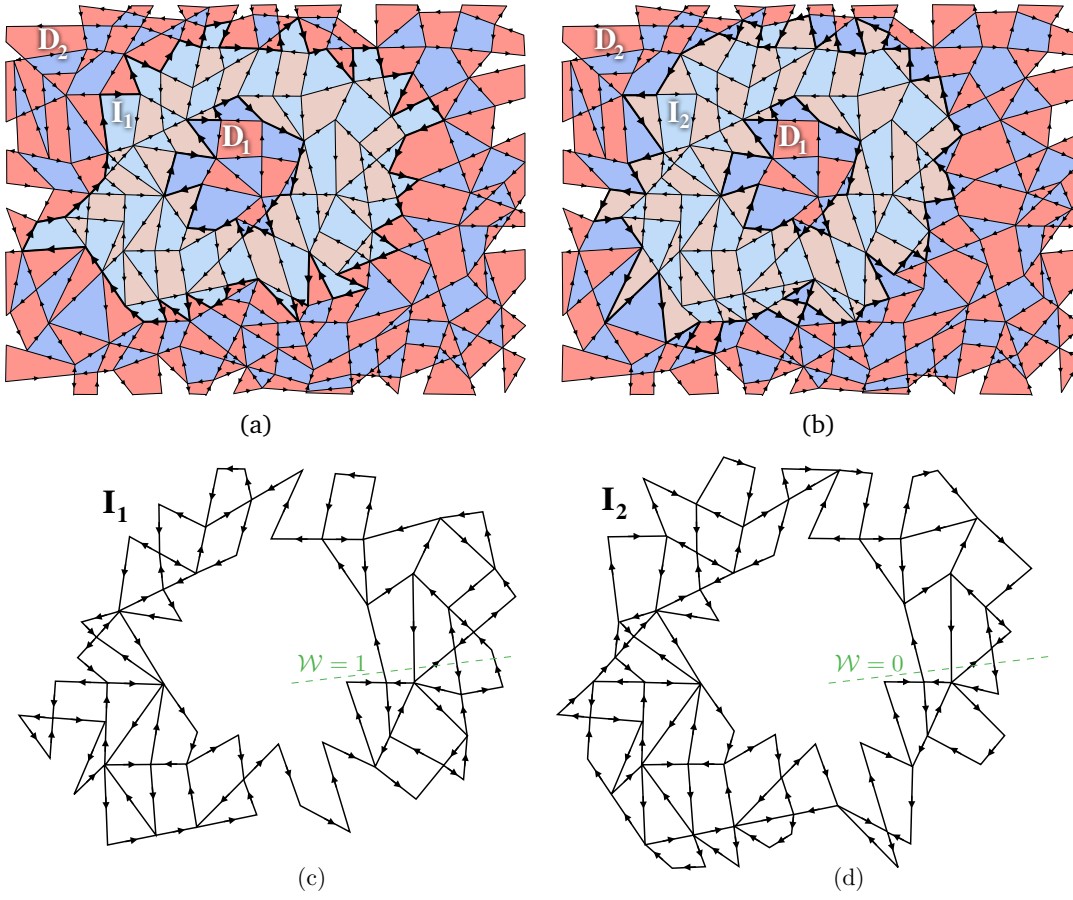

**Figure 8:** Eulerian graph $\mathcal{E}$. Each face is a assigned to a color or equivalently to an orientation (red + and blue −). (a) Circuits $\mathcal{C}_1$ and $\mathcal{C}_2$ have the same circulation (clockwise). (b) Circuits $\mathcal{C}_1$ and $\mathcal{C}_2$ have opposite circulation. The domain $D_1$ is the same in (a) and (b). (c) and (d): the two interface graphs $\mathbf{I}_1$ and $\mathbf{I}_2$ extracted from figures (a) and (b) respectively (and slightly enlarged for clarity) have different winding numbers.

Because of its corona shape, the interface graph **I** is not simply connected. An interface chiral mode would thus consist in a circuit in **I** that winds around $D_1$. In the following, we shall refer to the *circulation* of **I** as the property such that *any Eulerian circuit in* **I** *surrounds its central face*. One can thus ask under which condition the circulation of **I** is non-zero. The answer to this question only depends on the relative circulation of the delimiting contours $\mathcal{C}_1$ and $\mathcal{C}_2$ as follows (see Appendix C)

**Property 1.** *The circulation of* **I** *is opposite to the orientation of* $\mathcal{C}_1$ *and* $\mathcal{C}_2$ *when they are identical. Otherwise, the circulation of* **I** *is zero.*

In other words, when the two surrounding domains $D_1$ and $D_2$ have opposite circulation, an Eulerian circuit that winds around the central domain can be found in the interface graph. In general, this circuit is not unique, but whatever its peculiar sinuosity, it has to wind around $D_1$ with the same fixed orientation. There is no Eulerian circuit that does not wind around $D_1$. This is the case of $\mathbf{I}_1$ in figure 8 (c) that has been extracted from the figure 8 (a). In contrast, an Eulerian circuit such as $\mathbf{I}_2$ depicted in figure 8 (d), for which $D_1$ and $D_2$ have the same orientation, cannot wind around $D_1$. Two examples of such Eulerian circuits are shown in figures 9 (a) and (a'). Note that in the second case, it may be possible to find a winding circuit which is not Eulerian. In that case, other circuits can be found with the opposite orientation so that the total vanishing circulation of the graph is preserved. This is illustrated in figure 9 (b').

More generally, any decomposition of the interface graph must preserve the circulation. In particular, when the circulation vanishes, a minimal Veblen decomposition can be found, thus preventing the existence of any winding oriented graph, as illustrated in figure 9 (d'). In contrast, a non-zero circulation prevents the minimal Veblen decomposition: in addition to the disconnected cycles, any decomposition into elementary cycles yields a winding circuit with a fixed orientation, as displayed in figures 9 (c) and (d).

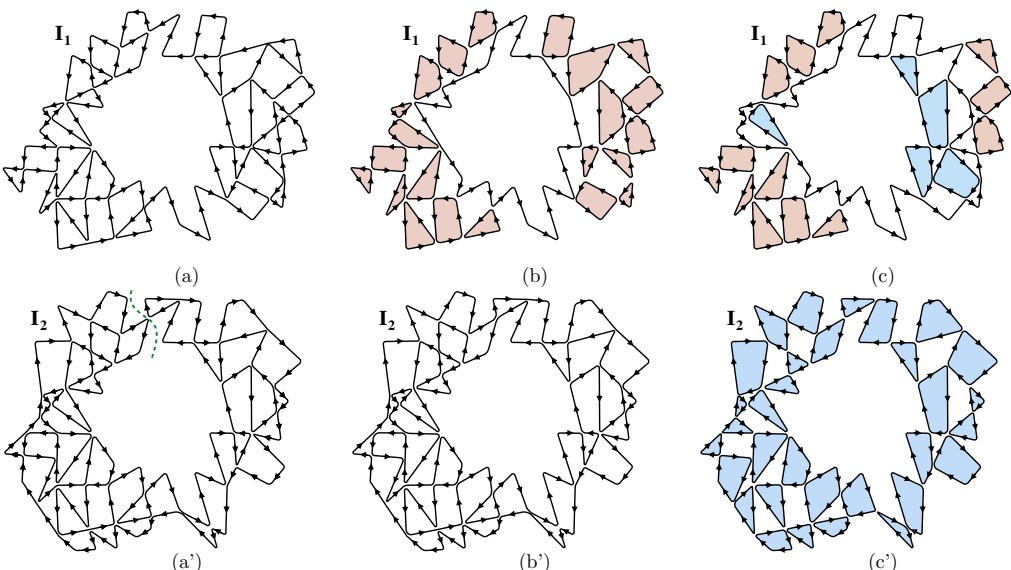

**Figure 9:** Interface graph $\mathbf{I}_1$ (top) and $\mathbf{I}_2$ (bottom). (a) and (a') are Eulerian circuits that wind and does not wind respectively around the central face. (b') decomposition of $\mathbf{I}_2$ into a clockwise and a counter clockwise circuits. (b) and (c) are examples of non-minimal Veblen decompositions"of $\mathbf{I}_1$. (c') is the minimal Veblen decomposition of $\mathbf{I}_2$.

This can be rephrased by the following rule

**Property 2.** *There exists a single oriented surrounding circuit in* $\mathbf{I}$
$$\Longleftrightarrow \mathbf{I} \text{ does not allow a minimal Veblen decomposition}$$

The Eulerian circuits discussed above are a particular case of such oriented surrounding circuit. This property allows one to simply evaluate graphically whether an arbitrary corona graph hosts a chiral circuit.

# 5  Topological aspects

## 5.1  Winding number of the graph

The existence of oriented surrounding circuits that prevent the minimal Veblen decomposition of the interface graph, like the Eulerian circuit, are other manifestations of an interface chiral mode. They cannot be reduced to a cycle around one of the face of the original interface graph. Instead they have to surround the interior of the corona. In that sense, they constitute non contractile loops. The next step is to encode the existence of these non contractile loops with winding number of the interface graph.

Let us define geometrically the winding number $\mathcal{W}$ of the graph $\mathbf{I}$ by the algebraic sum of the linking numbers given by the intersections of the oriented links of the graph with any semi-infinite line (not necessarily straight) starting from the central face of $\mathbf{I}$. Examples with $\mathcal{W} = 1$ and $\mathcal{W} = 0$ are shown in figures 8 (c) and (d) for $\mathbf{I}_1$ and $\mathbf{I}_2$. Following the lines of the appendix $C$, one can rephrase the property 1 as

**Property 3.** *There exists a (counter-)clockwise oriented surrounding circuit (i.e. chiral interface mode) in* $\mathbf{I}$  $\iff$  $\mathcal{W} = (-)1$

according to our sign convention. The absence of chiral interface mode is then encoded by $\mathcal{W} = 0$, which is satisfied if (and only if ) $\mathcal{C}_1$ and $\mathcal{C}_2$ have opposite orientations. Being defined as a purely geometrical quantity of the interface graph, $\mathcal{W}$ is invariant under the choice of scattering amplitudes at the nodes. In particular, it remains invariant for all the decompositions depicted previously: Eulerian circuits and Veblen decompositions. This winding number thus correctly accounts for the existence of an oriented interface mode whose existence is only fixed by the difference of orientations of the two surrounding domains.

Notice that, alternatively, a rotation number could also be defined to characterize the circuits in $\mathbf{I}$ that wind around the central face (see appendix B).

## 5.2  Quantized flow on the graph

In the previous section, we have characterized the circulation of a finite oriented Eulerian graph that constitutes the interface between two domains. The analysis was purely performed by means of elementary geometry and graph theory. All the graphs that were considered so far could be thought physically as the representations of non-periodic *classical* dynamics, since a state at a given link is fully transmitted to a single next link. This corresponds to very specific models, for which the scattering amplitudes at the nodes are chosen to be either 0 or 1.

This is of course not the case in general for a *coherent* dynamics, where the scattering amplitudes are complex numbers $|z| < 1$ that must only satisfy the unitarity of the scattering matrix at each vertex. However, we will show in that section that these simple graph properties derived in section 4.2 actually fully characterize the flow $\Phi$ that is entailed by the coherent dynamics occurring in the interface graph $\mathbf{I}$.

### 5.2.1  State $|\Psi\rangle$ on the graph

Let us label each link of the graph by an integer $n \in [1, N]$. A state $|\Psi\rangle$ of the system decomposes on the basis $\{|n\rangle\}$, and thus lives in the Hilbert space $\mathcal{H} = \ell^2(N; \mathbb{C})$. This state evolves through the step-evolution operator $\mathcal{U} \in U(N)$, similarly to (9), that describes the discrete dynamics on the finite oriented graph. $\mathcal{U}$ is obtained by interpreting each vertex of the graph as a unitary scattering process between $m$ incoming states $|in\rangle$ and $m$ outgoing states $|out\rangle$, *in* and *out* referring to the directions of the oriented edges that meet at the vertex. During the

evolution, the components of a state $|\Psi\rangle$ at time $t$ are transmitted at time $t + 1$ onto the adjacent oriented links according to the scattering coefficients at the nodes that preserve unitarity. Thus, in general, a state localized on a single edge at time $t$ is split at time $t + 1$ between the different outgoing edges of the node. This is what we mean by a *coherent dynamics* by opposition to a *classical dynamics* for which a state on a given link at time $t$ is fully transmitted to only one adjacent link at time $t + 1$, as in the previous section. Finally, one should stress that the step-evolution operator accounts for all the scattering processes occurring at all the nodes of the network simultaneously; if one thinks about a random graph as a network through which a wave packet spreads, one thus implicitly assumes a synchronisation for all the nodes to scatter simultaneously.

### 5.2.2 Flow operator

Let us introduce a flow operator $\hat{\Phi}$ that counts the net flow through a transverse section of **I** towards a region $P$, of the scattering amplitudes of a state that evolves according to the evolution operator $\mathcal{U}$. The flow $\Phi$ entailed by the coherent dynamics in the corona graph can then be defined as

$$\Phi \equiv \operatorname{tr}\hat{\Phi}. \tag{13}$$

Essentially, the flow operator $\hat{\Phi}$ is defined with a projector $\mathcal{P}$ that selects the half infinite region $P$ of the system toward which the flow is directed, and reads $\hat{\Phi} = \mathcal{U}^{-1}\mathcal{P}\mathcal{U} - \mathcal{P}$ [38, 45, 46]. When considering a flow along an *infinite* boundary, the trace of $\hat{\Phi}$ is an integer that corresponds to the number of boundary modes flowing along the edge. This integer was shown to be a topological index that enters the celebrated bulk-boundary correspondence, even in a dynamical regime where the system is periodically driven in time [19]. The expression of the flow above would however always yield zero in a finite graph **I** because of the periodicity of the corona along the edge. Indeed, the projector $\mathcal{P}$ necessarily defines two cross sections of **I**. As a consequence, every state that enters through one section will eventually exit through the other one, so that the two contributions cancel. In finite size systems, it is thus necessary to introduce a second projector $\mathcal{Q}$ in the definition of the flow, that restricts it through only one section of **I**, as depicted in figure 10 (a) [47]. A possible definition of the flow for a finite graph is then

$$\hat{\Phi} = \mathcal{Q}\mathcal{U}^{-1}\mathcal{P}\mathcal{U}\mathcal{Q} - \mathcal{Q}\mathcal{P}\mathcal{Q}. \tag{14}$$

At this stage, one needs to define more precisely the section through which the flow is defined. Any section must be a line that connects the interior of the corona to the exterior. As the links of the graph are associated to the basis elements of the Hilbert space, it is natural to impose the section line to avoid the links and thus to cross the nodes.

### 5.2.3 Quantized flow in the interface graph

Since each node of the section contributes additively to the flow, let us focus on a single node; the final result being straightforwardly inferred by summing the contributions of the different nodes on the cross section line. As sketched in figure 10 (b), let us denote by $N_{\mathcal{P}}^{in}$ the number of in-coming links to the node that are located in the region $P$ and by $N_{\mathcal{P}^\perp}^{in}$ the number of in-coming links that are located in the other side of the section. In the same way, let us denote by $N_{\mathcal{P}}^{out}$ (resp. $N_{\mathcal{P}^\perp}^{out}$) the number of out-going oriented links that leave the node in the region where $\mathcal{P}$ (resp. $\mathcal{P}^\perp$) applies. Therefore, unitarity imposes the conservation rule

$$N_{\mathcal{P}}^{in} + N_{\mathcal{P}^\perp}^{in} = N_{\mathcal{P}}^{out} + N_{\mathcal{P}^\perp}^{out}. \tag{15}$$

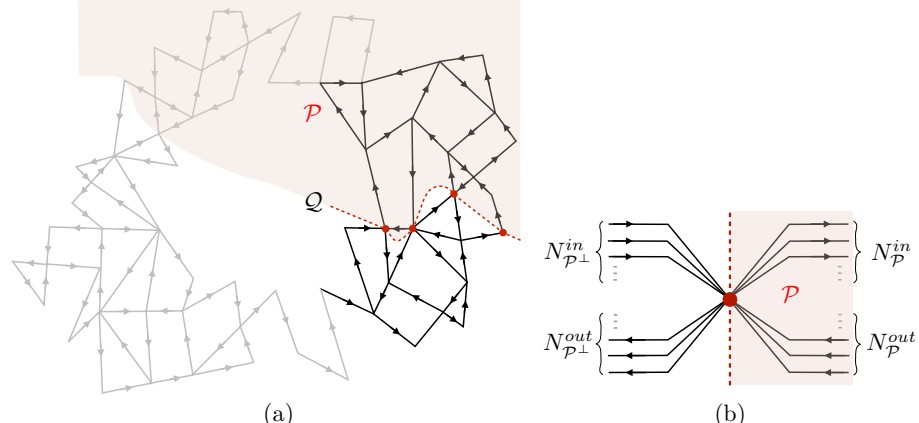

**Figure 10:** (a) Domains of application of the projectors $\mathcal{P}$ (light red colored background) and $\mathcal{Q}$ (thick graph). The flow is considered through the nodes of a cross section of the graph (dashed line), toward the light red domain. (b) Sketch of a node on the cross section.

The flow operator applies on a state $|\Psi\rangle$ that decomposes on a basis given by the links of the graph. Let us write $|p_i\rangle_{in}$ to refer to the base state $i \in [1, N_{\mathcal{P}}^{in}]$ localized on a link oriented toward a node of the section, in the domain where the projector $\mathcal{P}$ applies, and let us write $|p_i\rangle_{out}$ with $i \in [1, N_{\mathcal{P}}^{out}]$ in the case where the orientation of the link is reversed. In the same way, we shall write $|p_i^{\perp}\rangle_{in/out}$ to refer to a base state $i$ localized on a link in the domain where the projector $\mathcal{P}$ does *not* apply. The evolution operator on the graph around the section takes the unitary matrix form

$$\mathcal{U}\mathcal{Q} = \begin{pmatrix} R_{\mathcal{P}} & T_{\mathcal{P}\mathcal{P}^{\perp}} \\ T_{\mathcal{P}^{\perp}\mathcal{P}} & R_{\mathcal{P}^{\perp}} \end{pmatrix}, \tag{16}$$

where the block $R_{\mathcal{P}}$ encodes the reflection amplitudes in the domain where $\mathcal{P}$ applies, that is from states $|p\rangle_{in}$ to states $|p\rangle_{out}$, whereas $T_{\mathcal{P}\mathcal{P}^{\perp}}$ denotes the transmission amplitudes through the section from states $|p^{\perp}\rangle_{in}$ to states $|p\rangle_{out}$. Then one infers that

$$\mathcal{P}\mathcal{U}\mathcal{Q}|p_i\rangle_{in} = \sum_{j=1}^{N_{\mathcal{P}}^{out}} (R_{\mathcal{P}})_{ji} |p_j\rangle_{out} \tag{17}$$

and thus

$$\mathcal{Q}\mathcal{U}^{\dagger}\mathcal{P}\mathcal{U}\mathcal{Q}|p_i\rangle_{in} = \sum_{j=1}^{N_{\mathcal{P}}^{out}} (R_{\mathcal{P}})_{ji} \left[ \sum_{k=1}^{N_{\mathcal{P}}^{in}} (R_{\mathcal{P}}^{\dagger})_{kj} |p_k\rangle_{in} + \sum_{k=1}^{N_{\mathcal{P}^{\perp}}^{in}} (T_{\mathcal{P}\mathcal{P}^{\perp}}^{\dagger})_{kj} |p_k^{\perp}\rangle_{in} \right]. \tag{18}$$

The contributions to the trace of $\hat{\Phi}$ are given by ignoring the terms proportional to $|p_k^{\perp}\rangle_{in}$ and only keeping $k = i$ in the sum, so that one ends up with

$$_{in}\langle p_i | \hat{\Phi} | p_i \rangle_{in} \underbrace{=}_{\text{trace}} \left( \sum_{j=1}^{N_{\mathcal{P}}^{out}} |(R_{\mathcal{P}})_{ji}|^2 \right) - 1, \tag{19}$$

where the equality only holds for the terms that contribute to the trace. The same calculation can be carried out when considering now the incoming states $|p_i^{\perp}\rangle_{in}$ from the other side of the section, and one finds

$$_{in}\langle p_i^{\perp} | \hat{\Phi} | p_i^{\perp} \rangle_{in} \underbrace{=}_{\text{trace}} \sum_{j=1}^{N_{\mathcal{P}}^{out}} |(T_{\mathcal{P}\mathcal{P}^{\perp}})_{ji}|^2. \tag{20}$$

The trace of $\hat{\Phi}$ can be finally obtained by summing over all the $N^{in} = N^{in}_{\mathcal{P}} + N^{in}_{\mathcal{P}^\perp}$ incoming states $|i\rangle_{in}$ from the two sides of the section

$$\Phi = \sum_{i}^{N^{in}} \,_{in}\langle i| \hat{\Phi} |i\rangle_{in} \tag{21}$$

$$= \sum_{i}^{N^{in}_{\mathcal{P}}} \,_{in}\langle p_i| \hat{\Phi} |p_i\rangle_{in} + \sum_{i}^{N^{in}_{\mathcal{P}^\perp}} \,_{in}\langle p_i^\perp| \hat{\Phi} |p_i^\perp\rangle_{in} \tag{22}$$

$$= \sum_{j=1}^{N^{out}_{\mathcal{P}}} \left[ \sum_{i}^{N^{in}_{\mathcal{P}}} |(R_{\mathcal{P}})_{ji}|^2 + \sum_{i}^{N^{in}_{\mathcal{P}^\perp}} |(T_{\mathcal{P}\mathcal{P}^\perp})_{ji}|^2 \right] - N^{in}_{\mathcal{P}}, \tag{23}$$

where (19) and (20) have been inserted to get the last line. This expression simplifies by using the unitarity of $\mathcal{U}\mathcal{Q}$ (16), that implies

$$\sum_{i}^{N^{in}_{\mathcal{P}}} |(R_{\mathcal{P}})_{ji}|^2 + \sum_{i}^{N^{in}_{\mathcal{P}^\perp}} |(T_{\mathcal{P}\mathcal{P}^\perp})_{ji}|^2 = \sum_{i=1}^{N^{in}} |(\mathcal{U}\mathcal{Q})_{ji}|^2 = 1 \tag{24}$$

so that one finally gets

$$\Phi = N^{out}_{\mathcal{P}} - N^{in}_{\mathcal{P}}. \tag{25}$$

This result shows that the flow that is entailed by the coherent dynamics in the corona is quantized. Importantly, it does not depend neither on the choice of the cross section nor on the values of the scattering parameters that can thus be random. Instead, it is straightforwardly inferred by the structure of the graph itself. For instance, one can easily check that $\Phi = +1$ for the graph displayed in figure 10.

Remarkably, this integer number counts the circulation in the corona introduced in section 4.2, and as such, coincides with the winding number $\mathcal{W}$ of the graph **I**, that is

$$\Phi = \mathcal{W}. \tag{26}$$

This makes explicit the equivalence between the flow of the coherent dynamics on a graph and a topological property of the graph itself that can be obtained purely geometrically. Note that, consistently, these two quantities are defined by using an arbitrary line that crosses the graph.

Both definitions of the flow $\Phi$ and of the winding number $\mathcal{W}$ depend on a convention that fix their sign: the flow depends on which side of the cross section we choose to apply the projector $\mathcal{P}$, and the winding number is arbitrarily counted positively with the definition we gave in section 4.2, that gives $\mathcal{W} = +1$ when a loop is anti-clockwise oriented. Keeping this convention for $\mathcal{W}$, and choosing $\mathcal{P}$ to apply on the anti-clockwise side of the section, as we did above, allows us to finally write (26).

This analysis was led for an interface graph that is considered disconnected from the two domains $D_1$ and $D_2$. This is only achieved with special values of the scattering parameters, and in particular when $D_1$ and $D_2$ are in a Veblen decomposition. As in the cyclic case, where this decomposition corresponds to a strong phase rotation symmetric point, one can reasonably expect that the existence of the circulating chiral flow survives beyond this particular case, even though the quantized flow is certainly not strictly confined in the interface graph anymore, but may leak a bit into $D_1$ and $D_2$. One can also expect that a percolation transition in one of the two domains is a sufficient condition for the chiral flow to disappear.

# 6 Conclusion

We have shown the existence of chiral modes of topological origin in disordered scattering networks. The approach, based on graph theory, allows one to reduce the physical problem of the search for topological modes arising in two-dimensional unitary discrete-time dynamics (or quantum walk) to the simpler graphical ones assigned to Eulerian graphs, namely, Eulerian circuits, Veblen decompositions and the winding of a graph as defined in this paper. This reduction is motivated by the existence of a strong phase rotation symmetry that emerges in cyclic periodic oriented lattices at specific values of the scattering parameters and that yields anomalous Floquet topological interface states. Indeed, when the periodicity is recovered, we have explicitly shown the mapping from the discrete-time tight-binding models that are commonly used to investigate anomalous Floquet topological physics, to scattering networks models of the Chalker-Coddington type. In that sense, this work generalizes the concept of anomalous Floquet topological states to a class of dynamical systems where the periodicity in time is not required.

The graph approach also reveals a difference between chiral modes of a Chern phase and chiral modes of the anomalous one. The former are therefore somehow "more subtle" or "less obvious" so to speak, while, conversely, anomalous chiral states are therefore much simpler to implement in a given network (e.g. photonic or acoustic) as they can be engineered from a fully reflecting interface. In that perspective, the anomalous chiral states are therefore found to be rather common and ubiquitous in arbitrary unitary networks.

Besides, since the analysis is also based on finite graphs, it also does not require the notion of a bulk invariant, and can thus be applied to finite size systems of small size.

Finally, the approach applies for a given arbitrary (Eulerian) graph, so that the topological properties exist for a given configuration of disorder; there is therefore no need for an average over disorder configurations (in space or time) to recover the topological properties. From this perspective, topological disordered networks can thus somehow be seen as dynamical analogues of the recently introduced amorphous topological insulators [48].

Several directions seem natural for future works. Among them, it would be for instance interesting to investigate a continuous limit of these arbitrary Eulerian oriented graphs beyond the original Chalker-Coddington model [49], in order to explore the conditions for the existence of topological properties in non-periodic unitary dynamical quantum systems. Another question to be answered is the one of graphs that would yield more than one chiral interface state. This seems to be achieved pretty naturally by allowing the graphs to be non *simple*, meaning that two nodes can be related by more than one oriented link. It seems straightforward to increase the value of the a winding number as $\mathcal{W} \to n\mathcal{W}$ by multiplying each oriented link $n$ times. The question looks however more puzzling when this multiplication differs from link to link. Finally, all along this work, it was assumed a synchronization of the dynamics, meaning that all the components of a state on the graph, that live on the links, change simultaneously in time. We could imagine a less restrictive physical situation where a wave packet actually propagate in a graph whose links have different lengths so that the different scattered states reach the nodes at different times. Unlike the calculation of the quantized flow (25), the graph theory arguments developed in section 4.2 seem however independent of any synchronisation.

## Acknowledgements

I would like to acknowledge Jérémy Boutier for stimulating discussions on graph theory and Alain Joye, Joachim Asch, and Clément Tauber for illuminating discussions in particular about the flow operator.

**Funding information**   This work was supported by the French Agence Nationale de la Recherche (ANR) under grant Topo-Dyn (ANR-14-ACHN-0031).

## A  Construction of the phase rotation symmetry operator for cyclic lattices

The existence of loops here is reminiscent of the cyclic structure of the network, which originates itself from the periodic driving in the original Floquet tight-binding model. Different loops-configurations are obtained by considering the succession of links $a_j$ and $b_j$ in the unit cell. Exemples of such loops, going clockwise and anti-clockwise are shown in figure 6. Consider a domain of identical loops. The bulk evolution operator can always factorise as

$$\mathcal{U}(k) = \mathcal{B}(k)P, \tag{27}$$

where $\mathcal{B}(k)$ is a diagonal unitary matrix encoding the phases (including the Bloch phases) accumulated from one link to another, and where $P$ is a unitary matrix that owns one and only one coefficient 1 on each of its raws and each of its columns. Generically such matrices represent the elements of the permutation group (or symmetric group $S_N \in SU(N)$), which indeed captures the loop structure. One can always re-order the successive links of the loops by a change of basis $P_0 = MPM^{-1}$ so that

$$P_0 = \begin{pmatrix} 0 & \cdots & \cdots & 0 & 1 \\ 1 & 0 & \cdots & & 0 \\ & 1 & & & \vdots \\ & & \ddots & & 0 \\ 0 & \cdots & 0 & 1 & 0 \end{pmatrix}. \tag{28}$$

We notice that the eigenvalues of $P_0$ are the $N$ roots of unity $\{\lambda, \lambda^2, \cdots, \lambda^N\}$ with $\lambda = e^{i2\pi/N}$. From there, one defines the diagonal matrix $\Lambda_0$ that lists these eigenvalues as

$$\Lambda_0 = \mathrm{diag}\left(\lambda, \lambda^2, \cdots, \lambda^N\right). \tag{29}$$

It follows that

$$\Lambda_0 P_0 = \begin{pmatrix} 0 & \cdots & \cdots & 0 & \lambda \\ \lambda^2 & 0 & \cdots & & 0 \\ \vdots & \lambda^3 & & & \vdots \\ & & \ddots & & 0 \\ 0 & \cdots & 0 & \lambda^N & 0 \end{pmatrix} \tag{30}$$

and also

$$(P_0 \Lambda_0)^{-1} = \begin{pmatrix} 0 & \lambda^{-1} & \cdots & & 0 \\ \vdots & 0 & \lambda^{-2} & \cdots & 0 \\ & & & \ddots & \ddots & \\ & & & & 0 & \lambda^{1-N} \\ \lambda^{-N} & 0 & \cdots & 0 & 0 \end{pmatrix} \tag{31}$$

so that

$$\Lambda_0 P_0 (P_0 \Lambda_0)^{-1} = \lambda \operatorname{Id} \tag{32}$$

and thus

$$\Lambda P (P\Lambda)^{-1} = \lambda \operatorname{Id} \tag{33}$$

after the change of basis

$$\Lambda \equiv M^{-1}\Lambda_0 M . \tag{34}$$

The matrices $\mathcal{B}(k)$ and $\Lambda$ being diagonal, they commute, so that one infers from (33) that the evolution operator associated to this configuration of loops satisfies

$$\Lambda \mathcal{U}(k)\Lambda^{-1} = \lambda \mathcal{U}(k) . \tag{35}$$

This is precisely the phase rotation symmetry defined in Eq (10). It follows that the $N$ bands of $\mathcal{U}(k)$ carry a vanishing first Chern number, as long as the bands do not cross when varying the parameters $\theta_j$'s away from the critical value for which the loop configuration is obtained. This result does not depend on the parity of $N$ (while $N$ was implicitly assumed even in [34]). This reenforces the link between the loops and the phase rotation symmetry, as we have an explicit expression of the operator $\Lambda$ in terms of the eigenvalues of $P_0$ and the shape of the loops encoded into $P$.

The reasoning above not only relates the loops configuration to the phase rotation symmetry, but also shows how to obtain an explicit expression of the phase-rotation symmetry operator in a concrete situation. As an example, consider the two domains of disconnected loops represented in figure 6 in the L-lattice. They are defined by blue clockwise cycles $c_1 : a_1 \rightarrow a_2 \rightarrow a_3 \rightarrow b_4 \rightarrow b_1 \rightarrow b_2 \rightarrow b_3 \rightarrow a_4$ and red counter-clockwise cycles $c_2 : a_1 \rightarrow a_2 \rightarrow a_3 \rightarrow a_4 \rightarrow b_1 \rightarrow b_2 \rightarrow b_3 \rightarrow b_4$ respectively. These loops beeing made of 8 steps, this fixes $\lambda = e^{i\frac{2\pi}{8}}$. The factorization (27) of the bulk evolution operator (9) in each domain defines the permutation matrices $P$ associated to each domain of loops as

$$P_{c_1} = \begin{pmatrix} 0 & 0 & 0 & 0 & 0 & 0 & 1 & 0 \\ 0 & 0 & 0 & 0 & 0 & 0 & 0 & 1 \\ 1 & 0 & 0 & 0 & 0 & 0 & 0 & 0 \\ 0 & 1 & 0 & 0 & 0 & 0 & 0 & 0 \\ 0 & 0 & 1 & 0 & 0 & 0 & 0 & 0 \\ 0 & 0 & 0 & 1 & 0 & 0 & 0 & 0 \\ 0 & 0 & 0 & 0 & 0 & 1 & 0 & 0 \\ 0 & 0 & 0 & 0 & 1 & 0 & 0 & 0 \end{pmatrix} \qquad P_{c_2} = \begin{pmatrix} 0 & 0 & 0 & 0 & 0 & 0 & 0 & 1 \\ 0 & 0 & 0 & 0 & 0 & 0 & 1 & 0 \\ 1 & 0 & 0 & 0 & 0 & 0 & 0 & 0 \\ 0 & 1 & 0 & 0 & 0 & 0 & 0 & 0 \\ 0 & 0 & 1 & 0 & 0 & 0 & 0 & 0 \\ 0 & 0 & 0 & 1 & 0 & 0 & 0 & 0 \\ 0 & 0 & 0 & 0 & 1 & 0 & 0 & 0 \\ 0 & 0 & 0 & 0 & 0 & 1 & 0 & 0 \end{pmatrix} . \tag{36}$$

The matrix $M$, that accounts for the change of basis, can then be obtained as $M = BA^{-1}$ where $A$ and $B$ satisfy $P = A\Lambda_0 A^{-1}$ and $P_0 = B\Lambda_0 B^{-1}$. These matrices are easily found explicitly

as

$$B = \begin{pmatrix} \lambda^8 & \lambda^8 & \lambda^8 & \lambda^8 & \lambda^8 & \lambda^8 & \lambda^8 & \lambda^8 \\ \lambda^7 & \lambda^6 & \lambda^5 & \lambda^4 & \lambda^3 & \lambda^2 & \lambda & \lambda^8 \\ \lambda^6 & \lambda^4 & \lambda^2 & \lambda^8 & \lambda^6 & \lambda^4 & \lambda^2 & \lambda^8 \\ \lambda^5 & \lambda^2 & \lambda^7 & \lambda^4 & \lambda & \lambda^6 & \lambda^3 & \lambda^8 \\ \lambda^4 & \lambda^8 & \lambda^4 & \lambda^8 & \lambda^4 & \lambda^8 & \lambda^4 & \lambda^8 \\ \lambda^3 & \lambda^6 & \lambda & \lambda^4 & \lambda^7 & \lambda^2 & \lambda^5 & \lambda^8 \\ \lambda^2 & \lambda^4 & \lambda^6 & \lambda^8 & \lambda^2 & \lambda^4 & \lambda^6 & \lambda^8 \\ \lambda & \lambda^2 & \lambda^3 & \lambda^4 & \lambda^5 & \lambda^6 & \lambda^7 & \lambda^8 \end{pmatrix}$$

$$A_{c_1} = \begin{pmatrix} \lambda^8 & \lambda^8 & \lambda^8 & \lambda^8 & \lambda^8 & \lambda^8 & \lambda^8 & \lambda^8 \\ \lambda^4 & \lambda^8 & \lambda^4 & \lambda^8 & \lambda^4 & \lambda^8 & \lambda^4 & \lambda^8 \\ \lambda^7 & \lambda^6 & \lambda^5 & \lambda^4 & \lambda^3 & \lambda^2 & \lambda & \lambda^8 \\ \lambda^3 & \lambda^6 & \lambda & \lambda^4 & \lambda^7 & \lambda^2 & \lambda^5 & \lambda^8 \\ \lambda^6 & \lambda^4 & \lambda^2 & \lambda^8 & \lambda^6 & \lambda^4 & \lambda^2 & \lambda^8 \\ \lambda^2 & \lambda^4 & \lambda^6 & \lambda^8 & \lambda^2 & \lambda^4 & \lambda^6 & \lambda^8 \\ \lambda & \lambda^2 & \lambda^3 & \lambda^4 & \lambda^5 & \lambda^6 & \lambda^7 & \lambda^8 \\ \lambda^5 & \lambda^2 & \lambda^7 & \lambda^4 & \lambda & \lambda^6 & \lambda^3 & \lambda^8 \end{pmatrix} \qquad A_{c_2} = \begin{pmatrix} \lambda^8 & \lambda^8 & \lambda^8 & \lambda^8 & \lambda^8 & \lambda^8 & \lambda^8 & \lambda^8 \\ \lambda^4 & \lambda^8 & \lambda^4 & \lambda^8 & \lambda^4 & \lambda^8 & \lambda^4 & \lambda^8 \\ \lambda^7 & \lambda^6 & \lambda^5 & \lambda^4 & \lambda^3 & \lambda^2 & \lambda & \lambda^8 \\ \lambda^3 & \lambda^6 & \lambda & \lambda^4 & \lambda^7 & \lambda^2 & \lambda^5 & \lambda^8 \\ \lambda^6 & \lambda^4 & \lambda^2 & \lambda^8 & \lambda^6 & \lambda^4 & \lambda^2 & \lambda^8 \\ \lambda^2 & \lambda^4 & \lambda^6 & \lambda^8 & \lambda^2 & \lambda^4 & \lambda^6 & \lambda^8 \\ \lambda^5 & \lambda^2 & \lambda^7 & \lambda^4 & \lambda & \lambda^6 & \lambda^3 & \lambda^8 \\ \lambda & \lambda^2 & \lambda^3 & \lambda^4 & \lambda^5 & \lambda^6 & \lambda^7 & \lambda^8 \end{pmatrix} \tag{37}$$

leading to the phase rotation symmetry operators

$$\Lambda_{c_1} = \mathrm{diag}(\lambda, \lambda^5, \lambda^2, \lambda^6, \lambda^3, \lambda^7, \lambda^8, \lambda^4) \tag{38}$$

$$\Lambda_{c_2} = \mathrm{diag}(\lambda, \lambda^5, \lambda^2, \lambda^6, \lambda^3, \lambda^7, \lambda^4, \lambda^8) \tag{39}$$

for the two disjoint domains.

## B  Rotation number

The circulation of a cycle could also be defined geometrically with the *rotation number*. There are several ways to evaluate the rotation number of an oriented polygon. One of them is to consider the sum of the deflection wedges $\theta_{j,j+1}$ between two consecutive links $j$ and $j+1$ (see figure 11 (a)), that is

$$\mathcal{R} = \frac{1}{2\pi} \sum_j \theta_{j,j+1} \quad \in \mathbb{Z}. \tag{40}$$

This number is an integer whose value may be different than $\pm 1$ if the polygon has self intersections, that is when the planar condition is released, as illustrated in figure 11 (b). Unlike the winding number, the rotation number of a closed loop does not depend on a choice of origin, and thus allows one to unambiguously define the orientation of the loop as the sign of $\mathcal{R}$, but would fail to capture some winding circuits in the case of non-planar graphs (that we do not consider in this work).

## C  Demonstration of Property 1

Let us split the property 1 into two parts as

**Property 4.** *The circulation of* **I** *is non-zero if and only if* $\mathcal{C}_1$ *and* $\mathcal{C}_2$ *have the same circulation.*

and

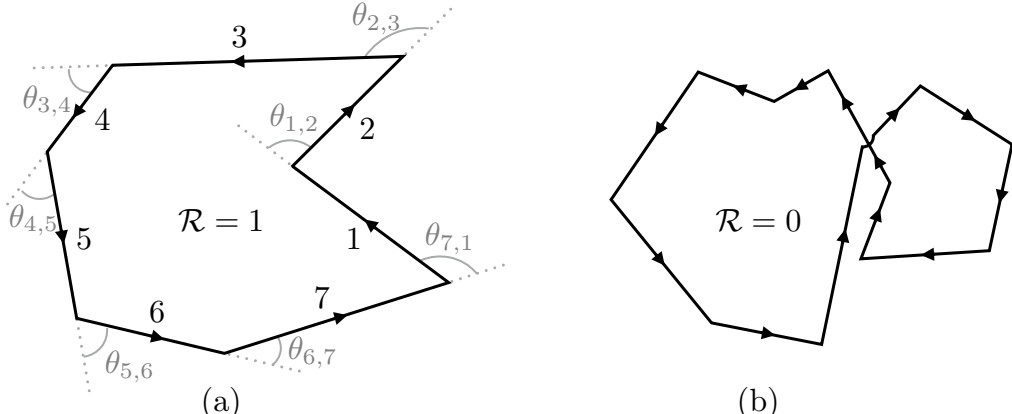

Figure 11: Oriented polygon without and with self-intersection (a) and (b). The deflection angles are depicted in gray.

**Property 5.** *when an Eulerian circuit in* **I** *does wind around* $C_1$*, its circulation is opposite to those of* $C_1$ *and* $C_2$*.*

To demonstrate the property 4, we use the Eulerian property of the total graph $\mathcal{E}$, by coloring each face with two colors. Then we notice that the adjacent faces to any circuit on $\mathcal{E}$ necessarily have the same color when they lie on the same side of the circuit. This is clearly shown in figure 8 where e.g. all the adjacent faces to $C_1$ that belong to $D_2$ are of type $+$ when the circulation of $C_2$ is clockwise (figure 8 (a)) and of type $-$ when it is anti-clockwise (figure 8 (b)).

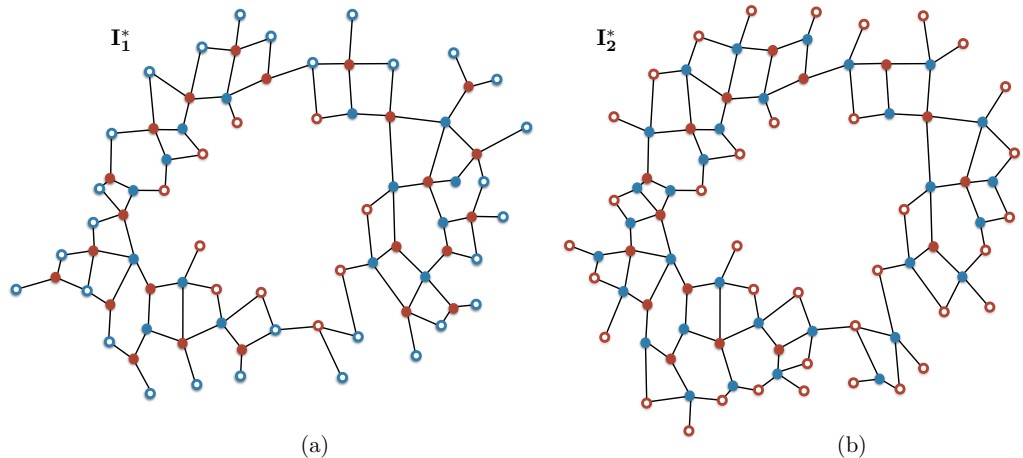

Figure 12: Dual graphs of $\mathbf{I}_1$ and $\mathbf{I}_2$. These graphs are bipartite since $\mathbf{I}_1$ and $\mathbf{I}_2$ are Eulerian. The interior and exterior nodes of the graphs are highlighted by colored rings rather than colored points.

This said, one has to distinguish two situations, depending on the circulations of $C_1$ and $C_2$, as illustrated in figure 8 (a) and (b).

First, consider that the circuits $C_1$ and $C_2$ have the same circulation. In that case, the outer adjacent faces to $C_1$ (namely the faces whose surrounding links belong to both $D_1$ and **I**) have a different color than the outer adjacent faces to $C_2$ (i.e. the faces whose surrounding links belong to both $D_2$ and **I**). In the example of figure 8 (a), the outer adjacent faces to $C_1$ are red whereas the outer adjacent faces to $C_2$ are blue. This provides an interface graph $\mathbf{I}_1$ that is represented alone in figure 8 (c) for clarity. Therefore, $\mathbf{I}_1^*$, the dual graph of $\mathbf{I}_1$, is delimited

by sites of different colors (see figure 12 (a)). As a consequence, any "transverse" walk $W$ in $\mathbf{I}^*$ that connects both sides of $\mathbf{I}_1^*$ has an odd length (odd number of links). This means that $W$ crosses an **odd** number of links of $\mathbf{I}_1$. It follows that any Eulerian circuit in $\mathbf{I}_1$ must cross $W$ an odd number of times, and therefore, it has to wind around $\mathcal{C}_1$. This proves the sufficient condition of the property 4.

The demonstration of the necessary condition can be obtained by showing that its contrapositive is true. Consider now that $\mathcal{C}_1$, and $\mathcal{C}_2$ *do not* circulate in the same direction (see figure 8 (d)). It follows that any "transverse" walk $W$ in $\mathbf{I}^*$ that connects both sides of $\mathbf{I}^*$ has now an even length. This is the case of $\mathbf{I}_2^*$, the dual graph of $\mathbf{I}_2$ (see figure 12 (b)), and thus, it crosses an even number of links of $\mathbf{I}_2$. We conclude that, in that case, any Eulerian circuit in $\mathbf{I}_2$ does not wind around $\mathcal{C}_1$.

This proves the property 4.

Next, to prove property 5, consider a transverse walk $W$ on $\mathbf{I}^*$ that starts from a site on the inner border and ends up at a site of the outer border. Then let us assign a "chirality" $+$ or $-$ to each step of $W$ in order to capture the orientation of the links of $\mathbf{I}$ that are crossed during the walk. More precisely, we could define this chirality as being given by the sign of the wedge product between a colinear vector to a link travelled on $\mathbf{I}^*$ during a step of $W$, with a colinear vector to the link that is crossed in $\mathbf{I}$. That way, the chirality of a transverse path on $\mathbf{I}^*$, given by the product of the chiralities of each of its steps, measures the unbalance between clockwise and counter-clockwise closed walks in $\mathbf{I}$. Importantly, the chirality of the transverse walks on $\mathbf{I}^*$ does not depend on the choice on the specific path across $\mathbf{I}$.

When $\mathcal{C}_1$ and $\mathcal{C}_2$ have an opposite circulation, then any transverse walk $W$ in $\mathbf{I}^*$ has a vanishing chirality, as expected, since the chiralities of the steps compensate pairwise. In contrast, when $\mathcal{C}_1$ and $\mathcal{C}_2$ have the same circulation, this number is non zero: if their rotation number is $-1$, then the chirality of any transverse walk on $\mathbf{I}^*$ that starts from a site on the inner border and ends up at a site of the outer border is $+1$ (as illustrated in figures 8 (a) and (c)), meaning that any Eulerian circuit on $\mathbf{I}$ circulates counter-clockwise.[2] In the same way, this weight is found to be $-1$ when the rotation number of $\mathcal{C}_1$ and $\mathcal{C}_2$ is $+1$, implying a clockwise circulation for any Eulerian circuits on $\mathbf{I}$. This proves the property 5.

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
