# Peer review of "Topological chiral modes in random scattering networks"

_SciPost Physics, doi:SciPost Phys. 8, 081 (2020)_

## Round 1 · Referee Report · Anonymous (Referee 1) · 2019-8-2

Strengths

Clear and well-written; elucidates and generalizes anomalous Floquet topological phases in a satisfying way.

Weaknesses

No serious weaknesses; a few small suggestions below in the Report.

Report

In this paper, the author maps the two-dimensional anomalous Floquet Anderson insulator (and related Floquet models) onto unitary scattering networks. Using graph theory and a “phase rotation symmetry” that such graphs obey, he gives a simple picture of when an anomalous phase (having a topological edge mode despite a vanishing Chern number) occurs in these models. This perspective allows him to generalize to non-Floquet evolution and unifies independent known results on scattering networks with known results in the Floquet setting. It also allows him to determine when an edge mode will occur even in cases where there is no known bulk invariant. This is a very nice paper, based around an insightful idea. I think it’s essentially publishable as-is, but I have a couple of minor points for the author:

  1. It would be good to comment on what happens to this picture with interactions. It seems at first glance that if the interactions are local, the scattering network map would still apply. Does this imply that the author’s graph-theoretic results also hold for the interacting case? Specifically does the argument in section 5.2 go through, and if not, why?

  2. On p8 the author writes, “According to the mapping established above, one can now interpret the topological transition between the Floquet anomalous and trivial regimes in discrete-time tight-binding models as a percolation transition.” The implications of this claim should be expanded. What are the universal features of the transition, if it maps onto percolation? Is this known in the literature? If not, it seems like an important result.

  3. There are a few small typos scattered throughout, so the paper should be given another close proofing. For instance, the labels (a) and (b) are missing in Fig 8.

I recommend the paper for publication.

  • validity: high
  • significance: high
  • originality: top
  • clarity: top
  • formatting: excellent
  • grammar: excellent

Author:  Pierre Delplace  on 2020-01-22  [id 715]

(in reply to Report 1 on 2019-08-02)
Category:
remark

1- The question of adding some interactions goes of course beyond the scope of this work, and it is in general a broad and difficult question. It honestly does not seem straightforward to me to claim any definitive sound statement about interactions at this stage, although I agree that this is a natural and stimulating direction.

I agree with the referee that local interactions are expected to only play a role at vertices, and thus will not totally ruin the construction proposed in this manuscript. It should be the case for instance if we think about the network as in the original Chalker-Coddington model, where the links correspond to semi-classical paths of wavepackets that eventually interact when they meet. My guess is that the existence of interface states should be robust to the introduction of interactions, as long as interactions do not change the Eulerian property of the graph (unitarity at the nodes) and the orientation of the links.

On the other hand one might also think that, due to interactions/non-linearities, an incident wavepackets is scattered to several wavepackets with different energies. This information is not visible in the present graphs, and might be relevant. Moreover, the quantization of the flow operator in section 5 essentially counts the number of uni-directional modes, in a single particle picture. The generalization to a many-body ground state is not straightforward, and would depend, I guess, on the kind of interactions we choose. For instance, a classical version (random walk) of the stepwise tight-binding model with hardcore repulsive interactions has been investigated in (Duncan et al., Driven topological systems in the classical limit, Phys. Rev. B 95, 125104, Ref [42] of the new version). It was shown that this classical interacting model maps the original quantum one exactly at the phase-rotation symmetric point, which shows another interest of this special point in the context of interactions. But this paper also shows that the directional current at the edge (in the case of a chiral edge state) depends on the filling factor in the problem, and more generally that the hardcore interactions strongly affect the directional edge current. So it seems that with interactions, at least in that case, the existence of unidirectional modes is not questioned, but the quantization of the current is.

2- This comment follows from an observation that was first made by Liang and Chong in their inspired paper PRL 2013 [Ref 35 of the resubmitted version]. They pointed out that, in the Chalker-Coddington (L-lattice) model, the gap closing point that accompanies the topological transition between anomalous and trivial regimes corresponds to a value theta=pi/4 of the scattering parameters, i.e. a 50/50 beam splitter at each vertex, which is precisely where the percolation transition is known to occur. The mapping I have established with the tight-binding model allows one to transfer this conclusion from the network model to the tight-binding model.

Note that this coincidence between the percolation transition and the topological transition seems to be a generic property: one can indeed easily check that for the Kagome and triangular networks as well, the topological transition between anomalous and trivial regimes comes together with an equal repartition of the scattering coefficients at the nodes, which is generically expected to correspond to a percolation transition. Accordingly, the quasi-energy spectrum is fully gapless at the transition, so that any state is free to propagate through the system (in the absence of gaps), by opposition to the phase rotation symmetric points where the bands are totally flat and the states thus cannot propagate. So generically, if one has a transition between an anomalous and a trivial regime, we know for sure that the spectrum must be fully gapless at some point, and therefore the states are free to propagate through the bulk, irrespectively of the randomness of the network in terms of number of links at each nodes (i.e. Within the framework of this study). Also, I am not claiming the reciprocal, that is that any percolation transition can be interpreted as a topological transition between an anomalous and a trivial regime.

Of course, I agree that a deeper analysis on the percolation transition in random networks and its link to topological transitions is an exciting direction of investigation that would deserve more work, and may for instance help revisiting the plateaus transitions in the original quantum Hall effect.

3- I thank the referee for pointing this out. Several typos have been corrected and the labels (a) and (b) have been added in Fig. 8.

---

## Round 1 · Referee Report · Anonymous (Referee 2) · 2019-8-31

Report

The paper deals with the topological nature of chiral modes appearing at interfaces in random oriented scattering networks. For certain cases, an exact mapping is found to Floquet topological insulators within the setting of non-interacting fermions. The paper presents an interesting perspective on the physics of topological edge/interface modes using graph theory, and their geometrical interpretation as such is undoubtedly of value. The paper is, for most part, clearly written and the exposition of the calculations is detailed well.

I recommend publication of the paper. However, I would like the author to address the following questions or discuss how they could be possibly answered.

  1. The analysis of the results depends quite crucially on the strong phase rotation symmetry and then the topological nature of the Chern numbers. It will be useful to have some idea about what happens as one tunes away from the symmetric point, even if the system stays in the same topological phase. An an example, in non-interacting femionic systems, the gap in the bulk spectrum can be related to the localisation of the chiral edge modes within the topological phase. How does something like this show up in the network picture?

  2. One of the strengths of the work presented here is that the presence of the chiral modes can be understood in non-periodic systems. Section 2 has a very clear description of how the cyclic oriented networks are obtained from the tight-binding models on regular lattices. I think the paper would benefit massively from a somewhat more detailed discussion of what kind of physical Hamiltonians and graphs should one think about when interpreting the results of the arbitrary disordered scattering networks.

  3. What changes in terms of the network picture between the anomalous Floquet topological phase (edge modes with zero Chern number) and the topological phase which is not anomalous?

  4. While it might be a topic of future research and outside the scope of this paper, it might be useful to have some directions about how the scattering network calculations need to be modified to take interactions into account.

Other than this, the paper, while generally clearly written, has quite a few typographical errors and would benefit from a thorough proof reading.

  • validity: -
  • significance: -
  • originality: -
  • clarity: -
  • formatting: -
  • grammar: -

Author:  Pierre Delplace  on 2020-01-22  [id 716]

(in reply to Report 2 on 2019-08-31)

1- The situation is pretty clear in the cyclic (regular) networks discussed in the paper, since they map onto discrete-time Floquet tight-binding models. The strong phase rotation symmetric point imposes the bands to be equidistant (in quasi-energy). In the models discussed here, all the bands then flatten (which is expected to be the generic case, because this flatness translates the localization of bulk modes that correspond to disconnected loops in the network picture, and therefore cannot propagate). When one tunes the scattering parameters away from this special point, the bands broaden by getting some dispersion so that all the gaps reduce. When decreasing the gaps, the localization length of the chiral modes increases (i.e. they spread more around the interface), until the gaps eventually close leading to the delocalization of the edge states. This is what one should qualitatively expect in the random case as well: when the scattering parameters of the two domains surrounding the interface network are tuned so that these domains cannot be seen as unions of disconnected loops, the chiral interface state starts spreading in each domain with a shape that one expects to be exponential. Of course the graph theory analysis is not quantitative in that respect, and a more quantitative study in terms of localization lengths and spectral gaps could be led numerically as a first step, by directly investigating the one-step evolution operator in random networks. It might be that the connectivity (number of links at each nodes) plays a role in the spreading of the chiral mode around the interface.

2- This is a good but difficult question. I tried unsuccessfully to provide a simple disordered version of the stepwise tight-binding models that I could map onto a random graph. In my opinion, this is an interesting open question: what kind of time or space disordered dynamics can one map onto an oriented graph (planar or not). An answer to that question may help bridging this work more clearly to Floquet Anderson insulators, even though I don’t expect the overlap to be strict, if only because random graphs are not supposed to describe periodically driven systems in time, as the referee stresses. Maybe a relevant direction to look at is more that of a time-dependent noise on top of a periodic drive. Anyway, random graphs are also interesting for themselves as they naturally describe the propagation of confined (acoustic or electromagnetic) waves in networks. In that perspective the scattering description is certainly more suitable than the Hamiltonian one.

3- The loops configurations, as the one shown in figure 6, from which the chiral mode is explicit, is specific to the phase rotation symmetric point, and thus to the anomalous regime. The (regular) chiral edge states of a Chern phase are not visible in this approach, since they cannot be seen as a classical path on the network. In other words, in a Chern phase, there is no special point in scattering parameter space at which all the scattering coefficients are either 0 or 1.

This remark was made in a previous work of ours with my collaborators M. Fruchart and C. Tauber (DOI 10.1103/PhysRevB.95.205413, Ref 34), and is in particular illustrated in figure 11 of that reference.

Thus, the graph approach reveals that chrial modes of a Chern phase behave differently from the anomalous ones in that respect. The former are therefore somehow more subtile or ‘’less trivial’’ so to speak. Conversely, anomalous chiral states are much simpler to implement in a given network for that reason.

4 - The question of adding some interactions goes of course beyond the scope of this work, and it is in general a broad and difficult question. It honestly does not seem straightforward to me to claim any definitive sound statement about interactions at this stage, although I agree that this is a natural and stimulating direction.

For instance one may expect local interactions to only play a role at vertices, and thus will not totally ruin the construction proposed in this manuscript. It should be the case for instance if we think about the network as in the original Chalker-Coddington model, where the links correspond to semi-classical paths of wavepackets that eventually interact when they meet. My guess is that the existence of interface states should be robust to the introduction of interactions, as long as interactions do not change the Eulerian property of the graph (unitarity at the nodes) and the orientation of the links.

On the other hand one might also think that, due to interactions/non-linearities, an incident wavepackets is scattered to several wavepackets with different energies. This information is not visible in the present graphs, and might be relevant. Moreover, the quantization of the flow operator in section 5 essentially counts the number of uni-directional modes, in a single particle picture. The generalization to a many-body ground state is not straightforward, and would depend, I guess, on the kind of interactions we choose. For instance, a classical version (random walk) of the stepwise tight-binding model with hardcore repulsive interactions has been investigated in (Duncan et al., Driven topological systems in the classical limit, Phys. Rev. B 95, 125104, Ref [42] of the new version). It was shown that this classical interacting model maps the original quantum one exactly at the phase-rotation symmetric point, which shows another interest of this special point in the context of interactions. But this paper also shows that the directional current at the edge (in the case of a chiral edge state) depends on the filling factor in the problem, and more generally that the hardcore interactions strongly affect the directional edge current. So it seems that with interactions, at least in that case, the existence of unidirectional modes is not questioned, but the quantization of the current is.

---

## Round 3 · Author Response

Dear editor,
I am glad that the two referees recommend this manuscript for publication. In this resubmitted version, I made a few slight improvements to clarify or stress some important points in the abstract and the conclusion, following the feedback I got from the referees, and also corrected missprints and typos. The two referees also both had questions about possible extensions of this work when including interactions. These fair but general questions go much beyond the scope of this work, and no modification of the manuscript has been made in that direction.

---

## Round 3 · List of Changes

. A sentence has been added to the abstract to stress the generalization of anomalous edge states beyond Floquet systems.
. After equation (1), \hbar has been put correctly at the denominator in the expression of theta.
. The expressions of the Floquet operator (6) and (8) and (9) have been corrected so that they correspond to the figure 1. This does not alterate any result, simply reflects a convention of the basis vectors of the lattice.
. Labels (a) and (b) have been added to figure 8.
. A piece of anwer has been given to one of the possible directions in the conclusion (the one concerning the higher values of the winding number).
. A sentence has been added in the conclusion that stresses the difference between edge states of a Chern phase and that of the anomalmous one.
. A more complete calculation has been provided in the derivation of the phase rotation symmetry operators in appendix A.

---

## Editorial Decision

published